# Comparing Forest Understory Fuel Classification in Portugal Using Discrete Airborne Laser Scanning Data and Satellite Multi-Source Remote Sensing Data

**Bojan Mihajlovski [1,2]** , **Paulo M. Fernandes [3,4]** , **José M. C. Pereira [5]** and **Juan Guerra-Hernández [5,*]**

1   FARMAHEM, Kicevska 1, P.O. Box 39, 1060 Skopje, North Macedonia; bojan.mihajlovski@lfmwb.net
2   Ministry of Agriculture, Forestry and Water Economy, Aminta Treti 2, 1000 Skopje, North Macedonia
3   Centre for the Research and Technology of Agro-Environmental and Biological Sciences, CITAB, University of Trás-os-Montes and Alto Douro, Quinta dos Prados, 5000-801 Vila Real, Portugal; pfern@utad.pt
4   ForestWISE—Collaborative Laboratory for Integrated Forest and Fire Management, Quinta de Prados, 5000-801 Vila Real, Portugal
5   Forest Research Centre, Associate Laboratory TERRA, School of Agriculture, University of Lisbon, Tapada da Ajuda, 1349-017 Lisboa, Portugal; jmcpereira@isa.ulisboa.pt
\*   Correspondence: juanguerra@isa.ulisboa.pt

**Abstract:** Wildfires burn millions of hectares of forest worldwide every year, and this trend is expected to continue growing under current and future climate scenarios. As a result, accurate knowledge of fuel conditions and fuel type mapping are important for assessing fire hazards and predicting fire behavior. In this study, 499 plots in six different areas in Portugal were surveyed by ALS and multisource RS, and the data thus obtained were used to evaluate a nationwide fuel classification. Random Forest (RF) and CART models were used to evaluate fuel models based on ALS (5 and 10 pulse/m$^2$), Sentinel Imagery (Multispectral Sentinel 2 (S2) and SAR (Synthetic Aperture RaDaR) data (C-band (Sentinel 1 (S1)) and Phased Array L-band data (PALSAR-2/ALOS-2 Satellite) metrics. The specific goals of the study were as follows: (1) to develop simple CART and RF models to classify the four main fuel types in Portugal in terms of horizontal and vertical structure based on field-acquired ALS data; (2) to analyze the effect of canopy cover on fuel type classification; (3) to investigate the use of different ALS pulse densities to classify the fuel types; (4) to map a more complex classification of fuel using a multi-sensor approach and the RF method. The results indicate that use of ALS metrics (only) was a powerful way of accurately classifying the main four fuel types, with OA = 0.68. In terms of canopy cover, the best results were estimated in sparse forest, with an OA = 0.84. The effect of ALS pulse density on fuel classification indicates that 10 points m$^{-2}$ data yielded better results than 5 points m$^{-2}$ data, with OA = 0.78 and 0.71, respectively. Finally, the multi-sensor approach with RF successfully classified 13 fuel models in Portugal, with moderate OA = 0.44. Fuel mapping studies could be improved by generating more homogenous fuel models (in terms of structure and composition), increasing the number of sample plots and also by increasing the representativeness of each fuel model.

**Keywords:** fuel mapping; LiDAR; ALOS-2 satellite; C-band SAR; sentinel; wildfires

## 1. Introduction

On average, vegetation fires burn around 760 Mha of land each year [1]. Although fires may start naturally, they are often caused by anthropogenic factors, with catastrophic consequences [2]. Indeed, wildfires are an important environmental problem in a wide range of global ecosystems [3]. Scientific projections for the future of climate change predict more intense and prolonged droughts in certain regions, followed by heavy rainfall and flooding events [4]. Such events are also becoming typical in the Mediterranean Basin, historically affected by intense wildfires that often result in large burned areas, having a significant impact on human lives [5,6].

Decision makers must carry out fire risk assessments to mitigate wildfires, including the identification of changes in fuel distribution, which requires time and skill and is costly [7]. Fuel types play a crucial role in the propagation of fires in ecosystems. Within the Mediterranean basin, fuel is derived from plant communities varying from shrubland to pine forests [8,9]. A previous analysis of fire selectivity in Portugal found that coniferous forests and shrublands are more prone to fire than agricultural areas [10,11]. In Portugal, fuel types are dominated by evergreen sclerophyll shrubs, which cover an area of about 1.6 million ha or 18% of the total area of Portugal [2]. In terms of wildfire mitigation, fuels can be treated to reduce fire hazards, which makes spatially explicit information and fuel mapping very important [12,13].

Following the development of fire behavior models, various fuel type classification systems have been created, e.g., Northern Forest Fire Laboratory–NFFL fuel models [14] or Canadian fuel types [15,16]. Due to the difficulties of assessing and mapping different fuel types, specific classification schemes are required for use in similar environmental conditions [17]. Considering similar spatial resolution and methodological approaches, fuel model assignment error is more likely to occur when a site-specific fuel classification system is used [18], compared to a standard classification system, e.g., NFFL. For Mediterranean areas, fuel models (UC040) are adapted fuel models revised by the US Forest Service [15] in the Andalucía (Spain) region [19]. For the purpose of fuel classification, a national classification system was developed in Portugal [13,20]. The system considers a matrix of percentage of litter and vegetation cover, resulting in four main fuel model groups. (L–litter, M–mixed, D–discontinuous and V–vegetation) associated with 18 fuel models [13].

Mapping fuel types is traditionally based on field surveys, which is challenging and expensive. However, remote sensing (RS) technology has made the entire process of mapping easier [15]. The ability to extract information from vertical and horizontal structural components makes LiDAR RS a very important tool for forest fuel mapping across large areas [18]. Various studies have addressed fuel type mapping on local, regional and global scales by using active and passive sensors [21–23]. Passive sensors cover a wide range of wavelengths within the spectrum, which makes the images acquired very useful for species identification and fuel classification [23]. However, passive sensors lack the ability to penetrate the canopy cover, and the data are therefore not suitable for describing forest fuel structure or understory vegetation composition [24]. Active sensors are therefore very important for monitoring forests on a global scale [25].

Two kinds of active sensors are used to map fuel types: LiDAR (Light Detection and Ranging), which uses light (in form of pulses) emitted from a laser, and RaDaR (Radio Detection and Ranging), which uses radio waves. As a result, LiDAR is used to estimate a variety of forest fuel variables, including canopy bulk density, canopy base height, canopy fuel load, and surface fuel metrics [26–32]. In terms of forest fuel mapping and the estimation of biomass, several RaDaR detection systems such as Airborne SAR (AIRSAR), GeoSAR, and Intermap Technology Corporation are available for spatial monitoring [33,34]. A recent study has shown that combining both passive and active sensors improves the results of fuel mapping in contrast to using only a single data source [35]. While multispectral passive sensors can be used to estimate species composition based on spectral response, LiDAR active sensors can extract the vertical forest structure, and therefore, the combined use of both types of sensors provides a novel and unique approach to mapping fuel types [7,36–38]. For instance, in a study undertaken in three different Mediterranean forests dominated by pines [17], satisfactory results were obtained when fuel types were mapped using ALS and Sentinel 2 data. Another study assessed the understory forest structure in combination with ALS and LANDSAT time series [39]. Most studies in the Mediterranean basin are based on the Prometheus Fuel Classification Scheme [17,40,41], but other classification schemes have also been used [37,42]. However, it is important to highlight that Prometheus Fuel Classification has the disadvantage of not being calibrated for local conditions [43].

Although fuel mapping in Mediterranean ecosystems has been previously addressed [17,44–49], there is a lack of research on the combined use of medium resolution multispectral and ALS data. To the best of our knowledge, no previous studies have developed simple, robust, and parsimonious models to provide forest managers with an alternative classification method for mapping Portuguese fuel models based on ALS data. On the other hand, very few studies have evaluated the impact of canopy cover and pulse density on the performance of fuel model classification in Mediterranean areas. Finally, this work aimed to develop the first fuel model classification in Portugal using C-band and L-band (SAR) backscatters, with multi-temporal Sentinel 2 and ALS data associated with a detailed field survey. In this regard, the goal of this study was to combine all sensor data, evaluate the performance of the models, and classify the fuel models according to the national fuel scheme within six study areas in Portugal.

The main objective of this research was to evaluate the potential of discrete return LiDAR data to classify fuels in six study areas in Portugal. The specific research objectives were as follows:

(1) Classify the four main fuel type groups in Portugal using Simple CART and RF models;
(2) Analyze the effect of canopy cover (CC) in Portugal on fuel classification accuracy;
(3) Investigate the performance of the models to classify the fuel types using different pulse densities (5 and 10 points m$^2$);
(4) Map fuel models by combining ALS data, Multispectral Satellite Imagery (S2) and C-L band SAR data (S1 and ALOS-2/PALSAR2 Satellite).

## 2. Data and Methods

### 2.1. Study Areas

ALS, satellite, and field data were acquired for the six target areas in Portugal, covering approximately 34,109 ha of land that was mostly occupied by vegetation but also included infrastructure and buildings. The most recent versions of the COS18 (Carta de Ocupação do Solo) used to extract the land use data corresponds to main fuel models of forest and agricultural areas https://www.dgterritorio.gov.pt/Carta-de-Uso-e-Ocupacao-do-Solo-para-2018 (accessed on 4 February 2022 (Figure 1)). The study areas include different environmental conditions and forest species distributions (Table 1). The elevation range within the study areas varies from 0 m.a.s.l. to 1203 m.a.s.l. (see Appendix A, Table A1).

### 2.2. Field Data

Field data were obtained within the framework of a pilot LiDAR project (áGiL TerFoRus-"Piloto sobre produtos de análise, com recurso a LiDAR, para a gestão do território, da floresta e dos fogos rurais") between April 2020 and June 2021 [50]. The set of áGiL TerFoRus training plots used to model fuel types consisted of 499 accurately georeferenced circular plots with a radius of 12.62 m (500 m$^2$). The plot locations were selected using a random sampling technique. Within each sample plot, four diagonal transects were defined at 45 degrees with a total of 33 points (8 points per diagonal) to describe fuel structure, allowing the classification of the fuel groups as per the fuel classification system used in Portugal. Therefore, for each plot, the mean height (h) and mean percent cover (C) of the litter, shrub, and herbaceous layers along the transects were calculated, and a fuel model from the Portuguese fuel model classification system was assigned (Table 2). The system comprises 18 fuel models (for more details and description see Appendix A, Table A2).

Field data resulted in 70 plots for fuel group D, 46 plots for F; 134 plots for M and 249 plots for fuel group V, further used as an input to run the classification. Due to the small number of observations and spectral similarity between the species of some fuel models, 13 fuel models were tested with the multi-sensor approach and a total of 475 field data plots (Table 3). The four fuel models for Mediterranean and Atlantic shrubland were merged into low shrubs (<1 m) and high shrubs (>1 m) models.

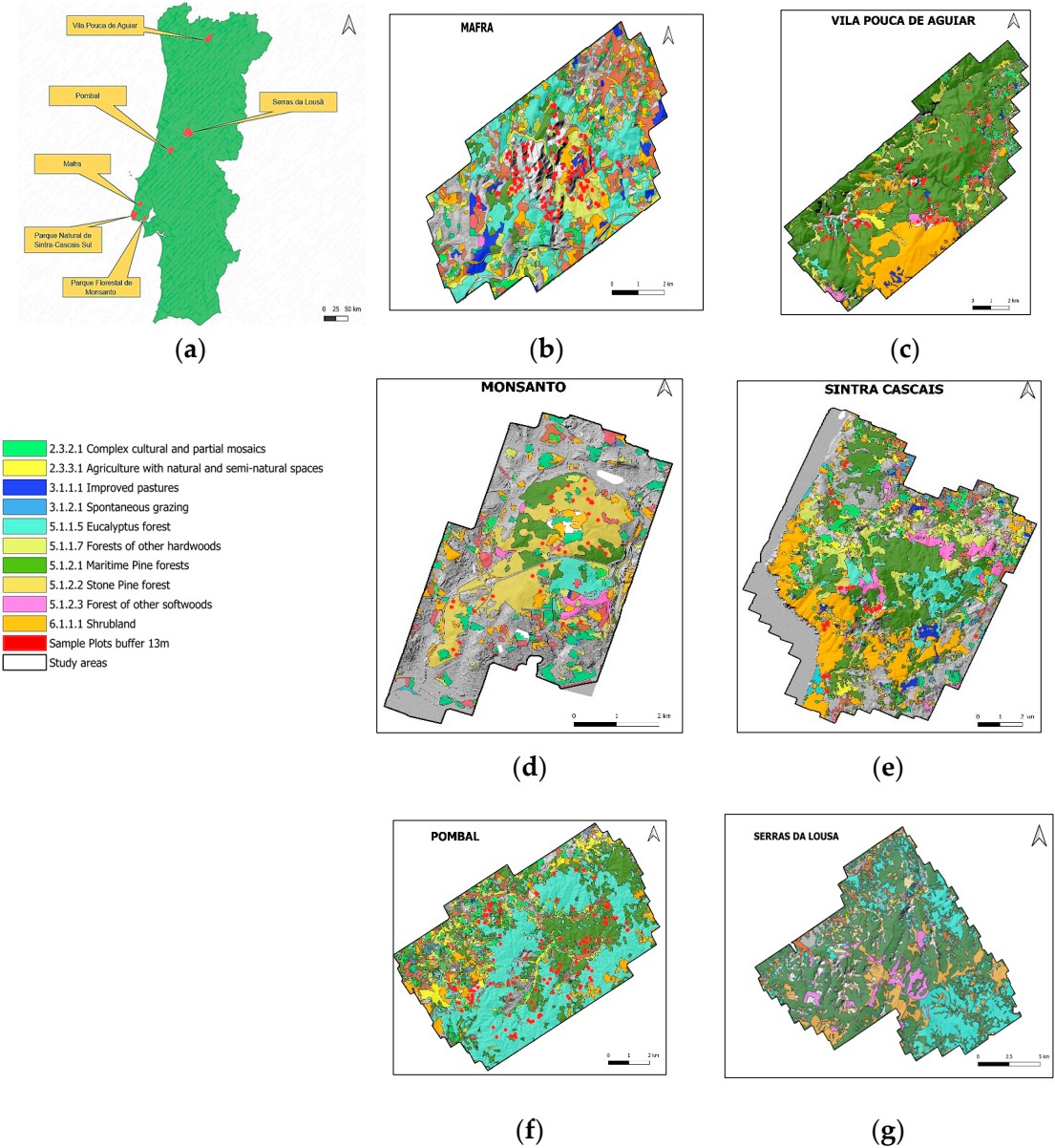

**Figure 1.** Locations of the six study areas in Portugal (**a**). Legend with the different colors represents the land use and land cover classification of agriculturas and forest classes for (**b**) Mafra, (**c**) Vila Pouca de Aguilar, (**d**) Monsanto, (**e**) Sintra Cascais, (**f**) Pombal, (**g**) Serras da Lousã. Red points represent the reference field plots.

**Table 1.** Species distribution for each study area (ha).

| Study Areas | *Eucalyptus globulus* | *Quercus suber* | *Pinus pinaster* | *Pinus pinea* | *Quercus* spp. |
|---|---|---|---|---|---|
| Mafra | 442 | 488 | 130 | 80 | --- |
| Pombal | 2251 | --- | 1073 | --- | 66 |
| Monsanto | 80 | --- | 159 | 498 | --- |
| Sintra Cascais | 655 | --- | 2111 | 192 | 31 |
| Serras da Lousã | 2428 | --- | 5212 | --- | 43 |
| Vila Pouca de Aguiar | 41 | --- | 3685 | --- | 259 |
| Total | 5897 | 488 | 12,370 | 770 | 399 |

**Table 2.** Portugal Fuel Classification Scheme [51] with the main four fuel type groups: D–Discontinuous fuel, F–Litter, M–Litter and Vegetation and V–Vegetation. C represents the cover of each fuel layer. Cover (C) and mean height (h) for each fuel layer.

| | VEGETATION; UNDERSTOREY; SHRUB OR GRASLAND | | | |
|---|---|---|---|---|
| LITTER | C < 1/3 | 1/3 > C < 2/3 | C > 2/3, h < 1 m | C > 2/3, h > 1 m |
| $C < \frac{3}{4}$ | D | D | V | V |
| $C < \frac{3}{4}$, h < 2 cm | F | M | M | V |
| $C < \frac{3}{4}$, h > 2 cm | F | M | M | M |

**Table 3.** Portuguese fuel models in the study areas.

| Fuel Type Group | Fuel Model Name | Number of Plots | No. per Fuel Models |
|---|---|---|---|
| F | F–EUC | 12 | |
| | F–FOL | 19 | 60 |
| | F–PIN | 29 | |
| M | M–CAD | 22 | |
| | M–ESC | 20 | |
| | M–EUC | 29 | |
| | M–EUCd | 15 | 173 |
| | M–PIN | 32 | |
| | M–F | 55 | |
| V | V–Ha | 42 | |
| | V–Hb | 35 | 242 |
| | V–Ma | 105 | |
| | V–Mb | 60 | |
| Total | | | 475 |

*2.3. ALS Data*

Publicly available ALS data for Portugal from the áGiL TerFoRus were provided by ICNF "https://geocatalogo.icnf.pt/geovisualizador/agil.html (accessed on 3 February 2022)". The data were acquired using Teledyne Galaxy PRIME Airborne LiDAR Terrain Mapper with SwathTRAK Technology. Average altitude in the first flight was 1250 m.a.s.l., and in the second one it was 1400 m.a.s.l. Sensors in the first flight used an average pulse rate of 900 kHz with a nominal outgoing pulse density of 10.02–13.88 points per m$^2$. The second flight used a pulse rate of 550 kHz with a pulse density of 5.13–6.56 points per m$^2$. Point clouds were provided from a total of 3210 tiles (1 km × 1 km coverage) in EPSG: 3763–ETRS89/Portugal TM06 Projection.

Standard ALS data processing was performed using LAStools software [52] and Notepad++ code editor. For each tile and sample plot, ALS-derived metrics (return percentiles, elevation statistics, canopy cover, coverage, density, and Canopy Relief Ratio) were extracted using a fixed height break threshold of 0.02 m for height metrics. The height break threshold, which is the limit for separating the point cloud data into two sets to separate canopy returns from the understory returns, was established at 4m (on the basis of field observations) for estimating canopy cover metrics (Table 4). V group represents the fuel model. For generation of the final DEM surface, the ALS ground points were converted to a TIN surface raster using 2 m raster DEM for all 6 study areas. A memory-efficient streaming technology was computed under three parallel processes using the *las2dem* command available in LAStools. As topographic predictor variables, Elevation, Slope and Aspect metrics were derived from the DEM data at 2 m spatial resolution, using QGIS 3.16.3 Hanover version [53].

A representation of main fuel groups modelled (F–Litter, M–Litter and Vegetation and V–Vegetation) was generated simulating the waveform from ALS point clouds (Figure 2). Using density histograms, the image illustrates the vertical distribution of point cloud height from the main fuel model groups. Litter (L) represents litter surface fuel layer

with slight density ALS returns from low interval height. Mixed (M) group is associated with a mix of litter with herbs, fens, or shrubs in the understory and the presence of trees. Mixed (M) group usually displays a concentration of ALS returns in more than one stratum. V (Vegetation) represents shrublands and grassland fuel models with a more even distribution of ALS returns and higher density of ALS returns concentrated in the lowest strata from the vertical distribution of vegetation. This condition could promote the vertical connectivity with the canopy fuels and how it is appreciated in the figure (i.e., an example of Mediterranean high shrubs with the presence of some trees).

**Table 4.** ALS metrics extracted from the data.

| ALS Metrics | Description |
| --- | --- |
| 1. HEIGHT METRICS | |
| 1.1. Central tendency of the ALS height distribution | |
| avg | Mean |
| 1.2. Dispersion of ALS height distribution | |
| std | Standard deviation |
| var | Variance |
| max, min | Maximum and minimum |
| 1.3. Shape of ALS height distribution | |
| ske | Skewness |
| kur | Kurtosis |
| 1.4. Percentiles of the ALS height distribution | |
| p01. p1..., p99 | 5th, 10th, 20th, 25th, 30th, 40th, 50th, ..., 90th, 95th, 99th percentiles |
| 2. CANOPY COVER METRICS | |
| 2.1. Canopy cover metrics | |
| Fixed Height Break Threshold) 4 m | |
| Cov | Percentage of first returns above HBT/total all first returns |
| Dns | Percentage of all returns above HBT/total all first returns |
| Crr | $=(h_{mean} - h_{min})/(h_{max} - h_{min})$ |
| 2.2. Relative Density and Vci metrics in bins height | |
| d00, d01, d02, d03, d04, d05, d06, d07, d08, d09 d10 | Relative density (number of returns divided by the total number of returns and scaled to percentage in this interval). |
| vc1, vc2, vc3, vc4, vc5, vc6, vc7 | Vertical complexity index (VCI) based on the diversity measurement indices that quantify vertical heterogeneity of the vegetation structure. |

Two structural diversity indices were calculated: Vertical Complexity Index (vci) and Relative Density metrics (d). Vci is an additional adaptation of the Shannon (H') index (see Equation (1)), which is sensitive to species diversity and species evenness. Values close to 1 indicate an equal number of LiDAR returns for most of the height bins. The VCI value decreases as the number of points per height bin increases (more uneven).

$$vci = (-\sum_{i=1}^{HB}[(p_i * \ln(p_i)]))/\ln(HB) \tag{1}$$

where HB is the total number of height bins, and $p_i$ is the proportional abundance of lidar returns in height bin i. Vci was computed for different vertical bin sizes of 0.20 m (vc1), 0.5 m (vc2), 1 m (vc3), 1.5 m (vc4), 2 m (vc6) and 10 m (vc7). In the case of relative height density metrics, the number of returns is divided by the total number of points and scaled to a percentage in the interval of each height bin; d00 (0–0.02 m), d01 (0.02–0.20 m),

d02 (0.2–0.5 m), d03 (0.5–1 m), d04 (1–1.5 m), d05 (1.5–2 m), d06 (2–4 m), d07 (4–10 m), d08 (10–20 m), d09 (20–30 m), d10 (30–40 m) were also computed.

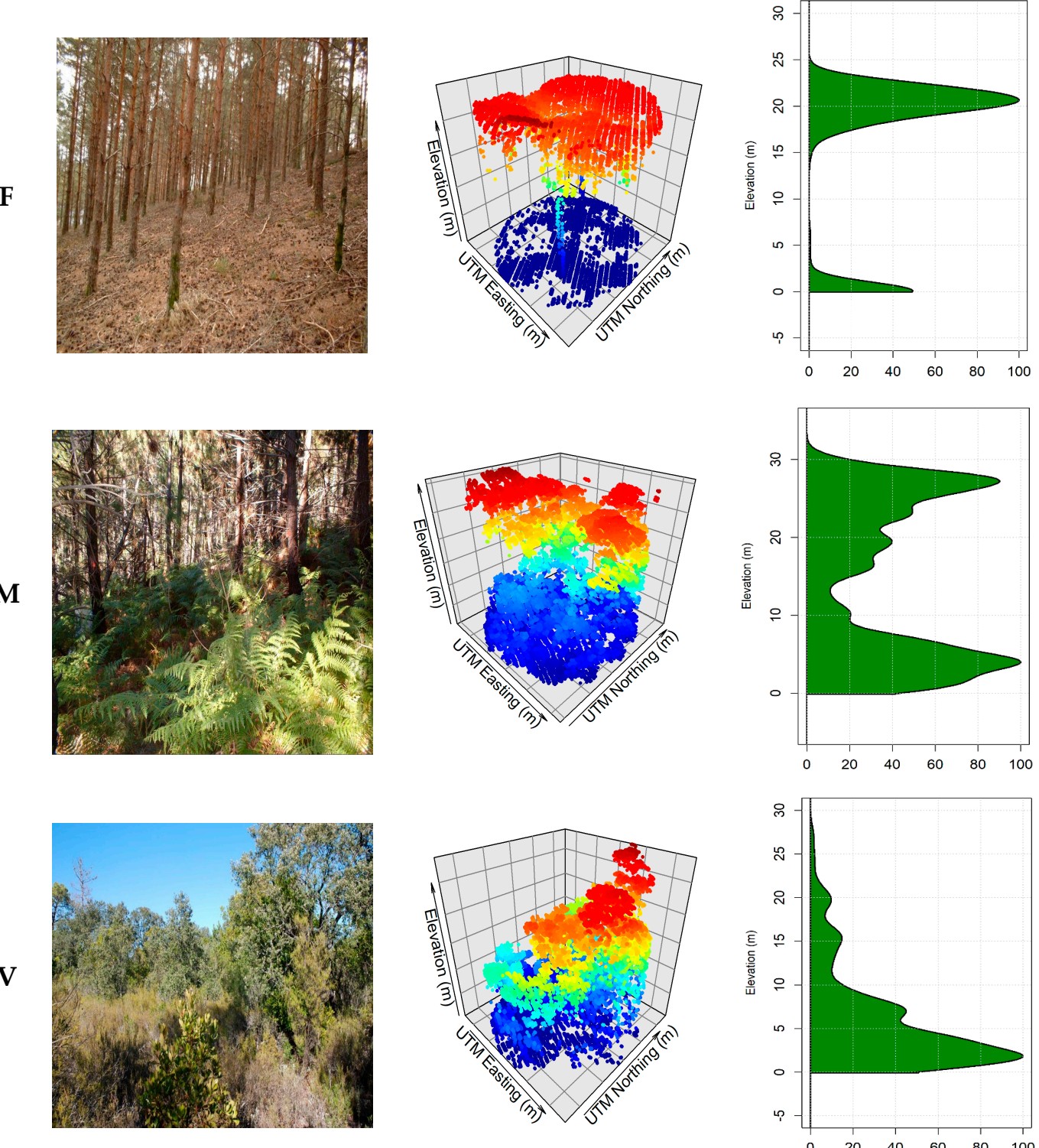

**Figure 2.** Examples of the main fuel groups modelled (F–Litter, M–Litter and Vegetation and V–Vegetation) in Portugal and associated extracted ALS point cloud and distribution of point cloud heights using rGEDI package [54] to simulate the waveform from ALS.

### 2.4. Multi-Spectral and SAR Data

Sentinel-2 images with less than 10% cloud cover over the six study areas were processed for summer, winter, and spring using Google Earth Engine (GEE) platform [55]. Pixel reflectance values were extracted from 10 bands of the Sentinel 2 (S2) using shapefile (.shp) with the location coordinates from each plot and each study area at a final resolution of 25 × 25 m (see Table A3, Appendix A). Four indices were then calculated: the normalized difference vegetation index (NDVI) [56], optimized soil adjustment vegetation index (OSAVI), green normalized difference vegetation index (GNDVI), and the normalized difference water index (NDVI) (Equations (2)–(5)).

$$NDVI = \frac{NIR\ (B8) - RED\ (B4)}{NIR\ (B8) + RED\ (B4)} \tag{2}$$

$$OSAVI = \frac{NIR\ (B8) - RED\ (B4)}{NIR\ (B8) + RED\ (B4) + L} * (L + 0.5)\ L\text{–Soil adjustment coefficient }(0.15) \tag{3}$$

$$GNDVI = \frac{NIR\ (B8) - RED(B3)}{NIR\ (B8) + RED\ (B3)} \tag{4}$$

$$NDVI = \frac{NIR\ (B3) - RED\ (B11)}{NIR\ (B3) + RED\ (B11)} \tag{5}$$

We obtained Sentinel-1 C band Synthetic Aperture RaDaR (SAR) images with horizontal (H) and vertical (V) polarization filtered for the period 1 June 2020 to 30 July 2020 at a spatial resolution of 10 m, later re-scaled to 25 m. The final S1 dataset was processed with the Gray Level Co-occurrence Matrix (GLCM) texture metrics [57]. Matrices were calculated for each image using the *glcm* library in R software to compute mean, variance and homogeneity metrics [58] (Table A3, Appendix A). To enable estimation of understory vegetation, the Phased Array L–band Synthetic Aperture RaDaR (PALSAR) satellite aboard the Advanced Land Observing Satellite (ALOS–2) was also used to calculate backscatter values in dual H and V polarization using yearly mosaic (Table A3, Appendix A). Two images in both polarizations were extracted at an original resolution of 25 m in WGS84/UTM zone 29 coordinate system. A total of 36 variables from S1 and S2, three topographical predictor variables from DEM (Elevation, Slope and Aspect), and 44 metric variables from the ALS sensor were used to classify fuel models using a multisensor approach.

### 2.5. Methods

A detailed flowchart is shown in Figure 3. Using already-produced metrics from ALS, S1, S2 and ALOS-2, fuel model classification was carried out using two different approaches. (i) A supervised learning approach, in which the Classification and Regression (CART) algorithm was used to fit a simple, robust model for the main four groups of fuel determined from ALS variables with the package "*rpart*" [59]. The number of folds of the cross-validation was set to 10 (*xval* parameter in *rpart* algorithm). CART models were obtained by pruning the tree using a complexity parameter (CP) implemented in the *rpart* library. The complexity parameter minimizes the number of splits at which the cross-validated classification error (*xerror* value- predicted residual error sum of squares (PRESS)) decreases and stabilizes. Then, a simple CART was obtained by selecting the number of terminal nodes in each tree in the cost-complexity pruning sequence (size value) (ii) Random Forest (RF) classifier implemented in package "*caret*" [60]. The RF classifier was optimized by tuning the hyperparameters *mtry* (the number of predicted variables randomly selected at each split) and *ntree* (number of trees) using the *expand.grid* function from the *caret* package.

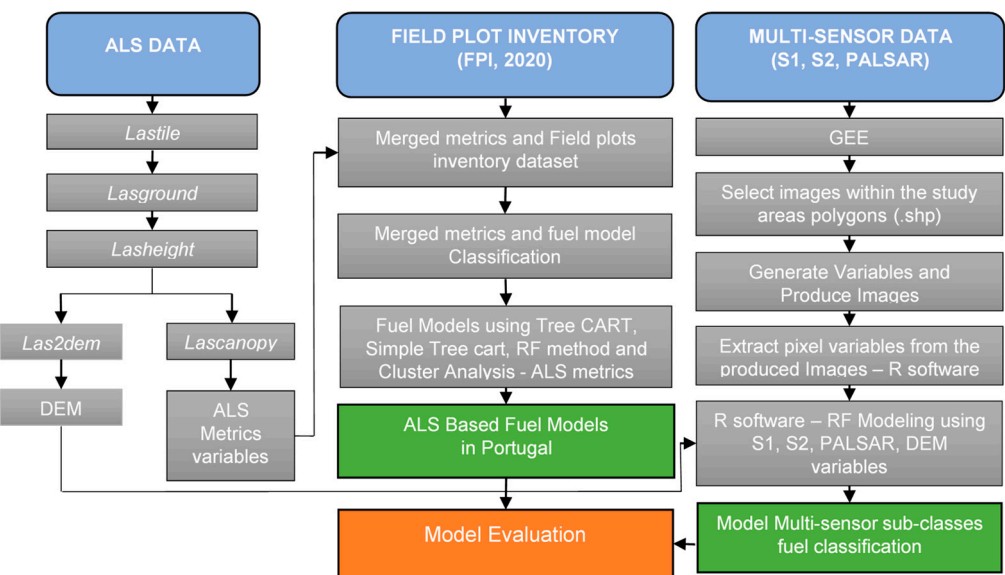

**Figure 3.** Flowchart based on Lastools software, scripts in R software and QGIS geospatial software.

Accuracy assessment for fuel type classification within the six study areas in Portugal was performed with the Confusion Matrix (CM), Producer (PA), User (UA) and Overall (OA) Accuracy metrics, and with the Kappa coefficient of agreement. Additionally, the importance variables indicator in R-studio was calculated from the RF model. Variable importance was tested by means of Mean Decrease Accuracy and Mean Decrease in Gini. Mean Decrease Accuracy works by measuring the decrease in the accuracy of the model when a particular feature is excluded from the dataset. It calculates the average reduction in accuracy across all decision trees in the forest for a given feature. Mean Decrease in Gini metrics, which measures how each variable contributes to the homogeneity of the nodes and leaves in the resulting RF [61]. In general, the mean decrease in the Gini coefficient increases with the importance of the variable in the model.

Finally, this study was guided by the following specific objectives: (i) classify the four main fuel type groups in Portugal using Simple CART and RF models; (ii) analyze the effect of canopy cover (CC) in Portugal on fuel classification accuracy. For this purpose, the Portugal National Forest Inventory (PNFI) was used for canopy cover classification to distinguish Sparse, Open and Dense Forest (Table 5); (iii) investigate the performance of the models to classify the fuel types using different pulse densities (5 and 10 points m$^2$).

**Table 5.** Canopy cover (CC) classes based on the NFI and ALS point dataset using different pulse densities (5 and 10 points m$^2$) with the number of observations per group and class.

| CC (%) | Description | Observations per Group | Observations per Class | | | |
|---|---|---|---|---|---|---|
| 10–30 | Sparse Forest | 147 | D:19 | F:7 | M:6 | V:115 |
| 30–60 | Open Forest | 135 | D:22 | F:0 | M:40 | V:71 |
| >60 | Dense Forest | 217 | D:29 | F:40 | M:87 | V:63 |
| **ALS Density (point/m$^2$)** | **Areas** | **Observations per Group** | **Observations per Class** | | | |
| 5 | Lousã, VPA, Pombal | 183 | D:32 | F:4 | M:60 | V:87 |
| 10 | Mafra, Sintra-Cascais, Monsanto | 316 | D:38 | F:42 | M:74 | V:162 |

## 3. Results

### 3.1. Fuel Classification Using ALS Data

The best three classification methods for the ALS approach are shown in Table 6. The CART produced better results than the RF model. The CART with cross validation (pruned at the minimum error) yielded an OA of 0.67 and a kappa of 0.45. Simple CART (pruned at 5-size tree) produced a fuel classification model with an OA of 0.61 and a kappa value of 0.39. The Random Forest classification yielded an OA of 0.60 and a kappa value of 0.34.

**Table 6.** Comparison of three classification methods using OA and Kappa values using only ALS variables.

| Four Main Fuel Classes (D, F, M, V) | | | |
|---|---|---|---|
| **Classification Method** | **OA** | **Kappa** | **Number of Plots** |
| CART with cross validation | 0.67 | 0.45 | |
| Simple CART | 0.61 | 0.39 | 499 |
| Random Forest (RF) | 0.60 | 0.34 | |

The CART with cross validation selected canopy cover (cov) at the first split, followed by relative density metrics—d02 at the second split (Figure 4). The possibility of classifying the D—discontinuous fuel model group is a crucial result. The Users Accuracy (UA) from the D and F fuel model groups achieved a UA 0.50 and 0.81, respectively.

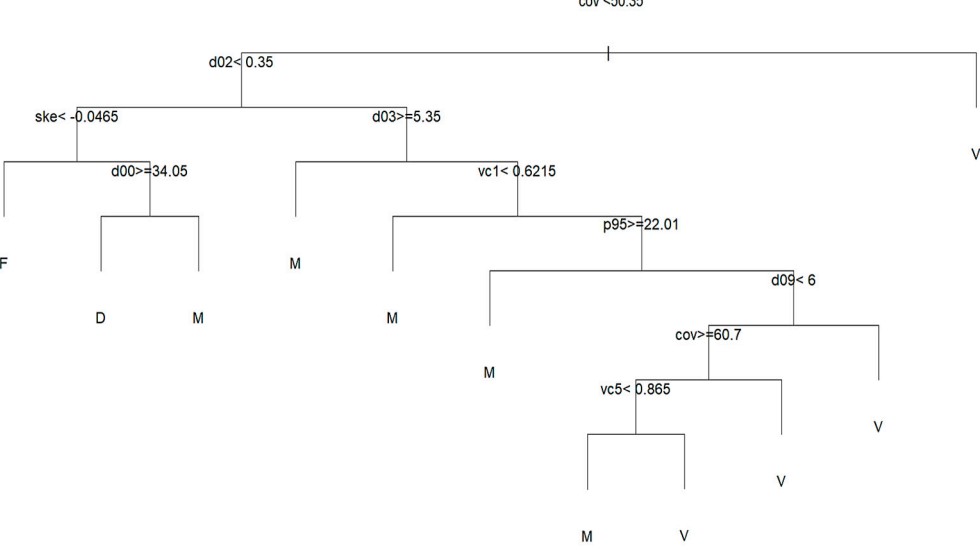

**Figure 4.** CART fuel classification model using ALS data (pruned tree at the minimum error (xerror)). Variables are described in Table 4.

However, in terms of Producers Accuracy (PA), CART classified fuel model group D with a PA of 0.11. The best results in terms of UA were obtained by fuel model groups V and M, with values of 0.70 and 0.63, respectively (Table 7, Figure 4).

### 3.1.1. Simple CART Model

The simple CART generates the simplest classification model and is easy to interpret (Figure 5). Pruned at the 5-tree size, the model has a very simple structure and has the possibility to classify a D fuel model group.

The canopy cover metric (cov) was the best variable to split the data between V and the rest of the fuel model groups. The relative density of points within the 0.2–0.5 m interval was used to separate the M and F group. Then, the skewness split between the F and D–M fuel model groups. Finally, the variable d00, which represents the percentage of coverage in the first 20 cm, was found to be the best predictor variable to split the data into M and

D fuel model groups. The confusion matrix (Table 8) correctly classified eight plots as D, twenty-seven as class F, ninety-seven plots as class M, and one-hundred and seventy as class V.

**Table 7.** CART with cross validation classification accuracy for the different fuel groups, where D = discontinuous fuel, F = Litter, M = Mixed, V = Vegetation, PA = Producer's accuracy, UA = user's accuracy, value shown in bold OA = overall accuracy and kappa value. Classification and reference (field check) frequencies are arranged in columns and rows, respectively.

| | Fuel group | Observed | | | | | |
| | | D | F | M | V | Σ | PA |
|---|---|---|---|---|---|---|---|
| Predicted | D | 8 | 1 | 15 | 46 | 70 | 0.11 |
| | F | 2 | 26 | 9 | 9 | 46 | 0.57 |
| | M | 3 | 3 | 88 | 40 | 134 | 0.66 |
| | V | 3 | 2 | 27 | 217 | 249 | 0.87 |
| | Σ | 16 | 32 | 139 | 312 | 499 | |
| | UA | 0.50 | 0.81 | 0.63 | 0.70 | OA | 0.68 |
| Kappa = 0.47 | | | | | | | |

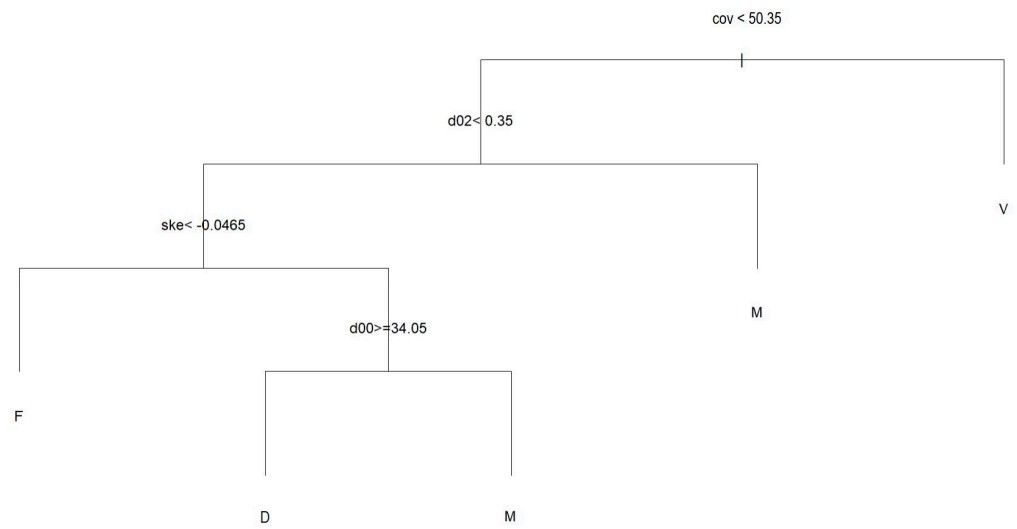

**Figure 5.** Simple CART fuel classification model using ALS data (pruned at the 5-tree size). Variables are described in Table 4.

**Table 8.** Simple CART classification accuracy for the different fuel groups, where D = discontinuous fuel, F = Litter, M = Mixed group, V = Vegetation, PA = Producer's accuracy, UA = user's accuracy, and value shown in bold OA = overall accuracy and kappa value. Classification and reference (field check) frequencies are arranged in columns and rows, respectively.

| | Fuel group | Observed | | | | | |
| | | D | F | M | V | Σ | PA |
|---|---|---|---|---|---|---|---|
| Predicted | D | 8 | 6 | 23 | 33 | 70 | 0.11 |
| | F | 2 | 27 | 11 | 6 | 46 | 0.59 |
| | M | 3 | 10 | 97 | 24 | 134 | 0.72 |
| | V | 3 | 8 | 68 | 170 | 249 | 0.68 |
| | Σ | 16 | 51 | 199 | 233 | 499 | |
| | UA | 0.11 | 0.59 | 0.72 | 0.68 | OA | 0.61 |
| Kappa = 0.39 | | | | | | | |

### 3.1.2. Random Forest Model

The RF model yielded an OA of 0.60 and a kappa value of 0.34. The RF model classified group D with a PA value of only 0.06. As an essential feature (similar to the previous method, pruned at the minimum error), the RF has the potential to estimate class D, although with low accuracy. The confusion matrix (Table 9) correctly classified four plots as D, nineteen as class F, seventy plots as class M, and one-hundred and ninety-two as class V.

**Table 9.** Accuracy of Random Forest classification of the different fuel groups, where D = discontinuous fuel, F = Litter, M = Mixed, V = Vegetation, PA = Producer's accuracy, UA = user's accuracy, and value shown in bold OA = overall accuracy and kappa value. Classification and reference (field check) frequencies are arranged in columns and rows, respectively.

|  | | **Observed** | | | | | |
|---|---|---|---|---|---|---|---|
| | Fuel group | D | F | M | V | Σ | PA |
| **Predicted** | D | 4 | 4 | 17 | 45 | 70 | 0.06 |
| | F | 3 | 19 | 13 | 11 | 46 | 0.41 |
| | M | 5 | 4 | 70 | 55 | 134 | 0.52 |
| | V | 10 | 4 | 43 | 192 | 249 | 0.77 |
| | Σ | 22 | 31 | 143 | 303 | 499 | |
| | UA | 0.18 | 0.61 | 0.49 | 0.63 | OA | 0.60 |
| Kappa = 0.34 | | | | | | | |

The contribution of variables such as cov (percentage of canopy cover), max (maximum dispersion of ALS height distribution), vc6 (vertical complexity index from bin size of 2 m), p95, and d02 (relative density of points between 0.2–0.5 m interval) were the most important variables in terms of Mean Decrease Accuracy (Figure 6). The most important variables for Mean Decrease Gini were cov (canopy cover), avg (mean central tendency of the ALS height distribution), d02, p70 and max (Figure 6).

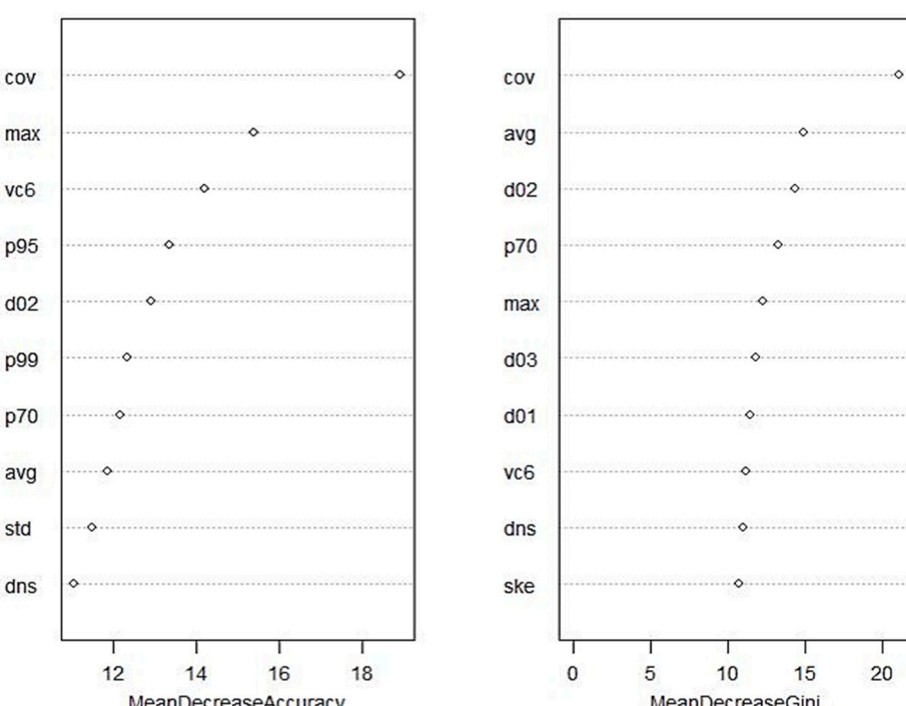

**Figure 6.** Ten most important variables from the RF model in terms of Mean Decrease Accuracy (**left**) and Mean Decrease in Gini (**right**).

### 3.2. Effect of Canopy Cover on Fuel Model Classification

The results of the effect of canopy cover on fuel model classification are shown in Table 10. In the case of Sparse Forest, it was not possible to fit a model using the cross-validation method due to the low number of observations per fuel model. Thus, CART without cross validation was used to generate a model, yielding an OA of 0.84 and a kappa value of 0.51. A simple CART model was used to prune the tree. In terms of accuracy, the simple CART yielded an OA of 0.82 and a kappa value of 0.35. The RF performed worse than the other models, with an OA of 0.76 a kappa value of 0.05.

**Table 10.** Results of Sparse, Open and Dense Forest Analysis.

| Method used | Sparse | | Open | | Dense | |
|---|---|---|---|---|---|---|
| | OA | Kappa | OA | Kappa | OA | Kappa |
| CART with-cross validation | No model | No model | No model | No model | 0.66 | 0.50 |
| CART without cross validation | 0.84 | 0.51 | 0.75 | 0.57 | 0.64 | 0.48 |
| Simple CART | 0.82 | 0.35 | 0.65 | 0.39 | 0.58 | 0.36 |
| Random Forest | 0.76 | 0.05 | 0.52 | 0.12 | 0.52 | 0.29 |

It was also not possible to fit a CART model with cross validation due to the small number of observations in Sparse and Open Forests. In the case of Open Forest, the CART model without cross validation generated the best results, with an OA of 0.75 and a kappa value of 0.57. The simple CART model (pruned at the 4-tree size) yielded an OA of 0.65 and a kappa value of 0.39. The RF model produced poor results, with an OA of 0.52 and a kappa value of 0.12. The final product shows the confusion matrix (see Appendix B; Table A5) of the best classified CART without the cross-validation model for the Open Forest analysis.

In the case of Dense Forest, the first model using CART with cross validation produced the best result, with an OA of 0.66 and a kappa value of 0.50. When the same model was used without cross validation, moderately good results were obtained, with an OA of 0.64 and a kappa value of 0.48. The simple CART model yielded an OA of 0.58 and a kappa value of 0.36. The RF model in the analysis again showed poor results, with an OA of 0.52 and a kappa value of 0.29. The confusion matrixes and CART models for the three types of forest are presented in Appendix B, Tables A4–A6 and Figures A1–A3.

### 3.3. Effect of ALS Pulse Density on Fuel Group Classification

Two ALS point densities were used in the analysis: 5 points and 10 points per $m^{-2}$. The performances of the models in terms of OA and kappa value are presented in Table 11.

**Table 11.** Results of the effect of using different ALS pulse densities.

| Method | 5 Points/m$^2$ | | 10 Points/m$^2$ | |
|---|---|---|---|---|
| | OA | Kappa | OA | Kappa |
| CART with cross validation | 0.61 | 0.33 | 0.71 | 0.51 |
| CART without cross validation | 0.71 | 0.53 | 0.78 | 0.65 |
| Simple CART | 0.67 | 0.42 | 0.69 | 0.49 |
| Random Forest | 0.51 | 0.19 | 0.68 | 0.48 |

For 5 points per $m^{-2}$, the CART model without cross validation produced the best result, with an OA of 0.71 and a kappa value of 0.53. The simple CART model yielded an OA of 0.67 and a kappa of 0.42. The RF model generated poor values, with an OA of 0.51 and a kappa value of 0.19. Table 12 shows the CM of the best estimated model CART without cross validation for 5 points per $m^{-2}$.

**Table 12.** CART without cross validation for 5 points per m$^{-2}$. Classification accuracy for the different fuel groups, where D = discontinuous fuel, F = Litter, M = Mixed, V = Vegetation, PA = Producer's accuracy, UA = user's accuracy, and value shown in bold OA = overall accuracy and kappa value. Classification and reference (field check) frequencies are arranged in columns and rows, respectively.

| | | Observed | | | | | |
|---|---|---|---|---|---|---|---|
| | Fuel group | D | F | M | V | Σ | PA |
| Predicted | D | 21 | 0 | 3 | 8 | 32 | 0.66 |
| | F | 1 | 0 | 1 | 2 | 4 | 0.00 |
| | M | 4 | 0 | 36 | 20 | 60 | 0.60 |
| | V | 9 | 0 | 5 | 73 | 87 | 0.84 |
| | Σ | 35 | 0 | 45 | 103 | 183 | |
| | UA | 0.60 | 0.00 | 0.80 | 0.81 | OA | 0.71 |
| Kappa = 0.53 | | | | | | | |

Class F, which was most problematic for estimation, was excluded from further analysis due to the low number of observations. Of these, 21 out of 32 were classified correctly as D class, 0 were classified as class F, 36 of 60 observations were classified as class M, and 73 of 87 observations were correctly classified as class V, with a higher PA of 0.84 in the analysis.

For the data's 10 points per m$^2$ density and the four main classes, the model provided accurate estimates. Detection of class D was problematic, with 11 out of 38 observations classified correctly; 30 of 42 observations were correctly classified as class F; 65 of 74 observations were classified as class M, with a PA of 0.88; and 140 of 162 observations were classified as class V, with a PA of 0.86 (Table 13).

**Table 13.** CART without cross validation for 10 points per m$^{-2}$ data. Classification accuracy for the different fuel groups, where D = discontinuous fuel, F = Litter, M = Mixed, V = Vegetation, PA = Producer's accuracy, UA = user's accuracy, value shown in bold OA = overall accuracy and kappa value. Classification and reference (field check) frequencies are arranged in columns and rows, respectively.

| | | Observed | | | | | |
|---|---|---|---|---|---|---|---|
| | Fuel Model | D | F | M | V | Σ | PA |
| Predicted | D | 11 | 3 | 9 | 15 | 38 | 0.29 |
| | F | 0 | 30 | 6 | 6 | 42 | 0.71 |
| | M | 0 | 2 | 65 | 7 | 74 | 0.88 |
| | V | 6 | 8 | 8 | 140 | 162 | 0.86 |
| | Σ | 17 | 43 | 88 | 168 | 316 | |
| | UA | 0.65 | 0.70 | 0.74 | 0.83 | OA | 0.78 |
| Kappa = 0.65 | | | | | | | |

### 3.4. Multi-Sensor Approach with Random Forest Classification

The performance of the RF model using combination of sets of variables from different sensors (ALS, S1, S2, PALSAR and DEM) is summarized in Table 14. The RF model, including all variables for 13 fuel model sub-groups in Portugal, produced an OA of 0.44 and a kappa value of 0.31.

**Table 14.** Overall accuracy and kappa value of RF model using multi-sensor approach.

| Data Matrix | OA | Kappa |
|---|---|---|
| ALS + S2 + DEM | 0.49 | 0.37 |
| ALS + S1 + S2 + PALSAR + DEM | 0.44 | 0.31 |
| ALS + DEM | 0.42 | 0.30 |
| ALS + S1 + PALSAR + DEM | 0.41 | 0.32 |
| S1 + S2 + PALSAR | 0.30 | 0.17 |

As shown in Table 14, the combination of optical satellite data from S2 and RaDaR data from S1 and PALSAR yielded the lowest values for both OA and kappa statistics. In all other combinations using ALS data, the results were similar and even close to those obtained with the best model. LiDAR matrix variables in combination with the optical variables derived from the S2 satellite produced the best results. Interesting findings were noted when the improvement in kappa value after the texture variables derived from the PALSAR and S1 SAR sensors were excluded.

The confusion matrix for the RF classification algorithm is shown in Table 15. Starting from the main fuel model groups, F was correctly classified in 17 of the 60 plots. Meanwhile, this group yielded one of the lowest mean PA values of 0.29. Group M achieved moderate results. Sixty-one of one-hundred and seventy-three plots were correctly classified, with a mean PA of 0.31. Group V produced the best results with this model, with 138 of 242 plots classified correctly. Furthermore, three fuel models were represented within the main fuel model group of litter (F). The model with the highest PA = 0.42 was F-EUC–Eucalypt litter. The Litter and Vegetation (M) fuel model group comprised six fuel models, two of which produced good results. M-ESC (evergreen hardwood litter and shrub understory) followed by M-F (litter with fern understory) produced the best results. Finally, group V (Vegetation) produced the best results in the model classification. The model estimates for four fuel models show great potential for classifying shrubs taller than 1 m (V-MAa) and shrubs lower than 1 m (V-MAb). Fuel model V-MAa with a PA of 0.69 was the better classified model in group V. The results also show that fuel model V-Ha (tall grass, >0.5 m) was more easily detected than V-Hb (low grass, <0.5 m), with a UA value of 0.57 and 0.37, respectively. Focusing on fuel model error in terms of structure revealed some similarities between classes. For instance, the model predicts M-F as V-MAa in 13 observations, whereas these two fuel models have similar structures, and only the type of understory vegetation varies. Another example is where the model predicts V-Hb as V-MAb in 11 observations, whereas those fuel models are similar and based on the <1 m height threshold. For instance, the fuel models with higher numbers of observations, such as V-MAa, with 105, produced the best results. Further, V-MAb has 60 observations, and again, the model easily detected those fuel structures. Furthermore, M-F had 55 observations, and V-Ha and V-Hb had 42 and 35, respectively.

**Table 15.** Multi-sensor approach ALS+S1+S2+PALSAR+DEM matrices using RF model; fuel model classification accuracy by group in the 13 fuel models considered, where PA = producer's accuracy, UA = user's accuracy, and value shown in bold OA = overall accuracy and kappa value. Classification and reference (field check) frequencies are arranged in columns and rows, respectively.

| | Fuel model | F-EUC | F-FOL | F-PIN | M-CAD | M-ESC | M-EUC | M-EUCd | M-F | M-PIN | V-Ha | V-Hb | V-MAa | V-MAb | Σ | PA | Mean PA per group | Fuel model error |
|---|---|---|---|---|---|---|---|---|---|---|---|---|---|---|---|---|---|---|
| | | | | | | | | Observed | | | | | | | | | | |
| Predicted | F-EUC | 5 | 0 | 2 | 0 | 0 | 4 | 0 | 0 | 0 | 0 | 0 | 1 | 0 | 12 | 0.42 | | 0.59 |
| | F-FOL | 0 | 3 | 1 | 2 | 3 | 0 | 0 | 6 | 0 | 0 | 0 | 4 | 0 | 19 | 0.16 | 0.29 | 0.84 |
| | F-PIN | 0 | 1 | 9 | 0 | 0 | 0 | 0 | 2 | 7 | 2 | 1 | 2 | 5 | 29 | 0.31 | | 0.69 |
| | M-CAD | 0 | 1 | 1 | 7 | 2 | 1 | 0 | 3 | 0 | 0 | 1 | 6 | 0 | 22 | 0.32 | | 0.68 |
| | M-ESC | 0 | 0 | 0 | 1 | 9 | 0 | 0 | 4 | 3 | 0 | 2 | 1 | 0 | 20 | 0.45 | | 0.55 |
| | M-EUC | 2 | 0 | 0 | 2 | 0 | 6 | 0 | 3 | 1 | 0 | 0 | 6 | 9 | 29 | 0.21 | 0.31 | 0.79 |
| | M-EUCd | 0 | 0 | 1 | 0 | 0 | 2 | 1 | 0 | 1 | 0 | 1 | 3 | 6 | 15 | 0.07 | | 0.93 |
| | M-F | 0 | 2 | 0 | 1 | 0 | 1 | 0 | 30 | 3 | 1 | 2 | 13 | 2 | 55 | 0.55 | | 0.45 |
| | M-PIN | 0 | 0 | 8 | 1 | 0 | 1 | 0 | 3 | 8 | 0 | 2 | 8 | 1 | 32 | 0.25 | | 0.75 |
| | V-Ha | 0 | 0 | 1 | 1 | 1 | 0 | 0 | 0 | 2 | 25 | 3 | 8 | 1 | 42 | 0.60 | | 0.40 |
| | V-Hb | 0 | 1 | 0 | 1 | 1 | 0 | 0 | 2 | 0 | 5 | 10 | 4 | 11 | 35 | 0.29 | 0.52 | 0.71 |
| | V-MAa | 0 | 2 | 0 | 0 | 2 | 4 | 0 | 12 | 2 | 6 | 2 | 72 | 3 | 105 | 0.69 | | 0.31 |
| | V-MAb | 1 | 0 | 2 | 0 | 0 | 1 | 1 | 4 | 1 | 5 | 3 | 11 | 31 | 60 | 0.52 | | 0.48 |
| | Σ | 8 | 10 | 25 | 16 | 18 | 20 | 2 | 69 | 28 | 44 | 27 | 139 | 69 | | | | |
| | UA | 0.63 | 0.30 | 0.36 | 0.44 | 0.50 | 0.30 | 0.50 | 0.43 | 0.29 | 0.57 | 0.37 | 0.52 | 0.45 | OA: | 0.44 | | |
| | Mean UA per group | | | 0.43 | | | | 0.41 | | | | | 0.48 | | | | | |
| | Kappa | 0.31 | | | | | | | | | | | | | | | | |

Figure 7 shows the most important independent variables for the model. Most of the variables were derived from ALS data, starting from the first and the most important d05 variable (1.5–2 m interval). The results indicate that the most important variables are those in the height interval from 0.20 to 1.5 m (d02, d03, d04) and cov, while the optical-derived variable is B9_Spring. Mean Slope and Altitude were two of the most important

topographic variables derived from the DEM. In general, the most important variables were (1) derived from the ALS sensors, (2) delivered from optical sensors (S2), and (3) the S1/PALSAR satellite sensor (Figure A4 in Appendix B).

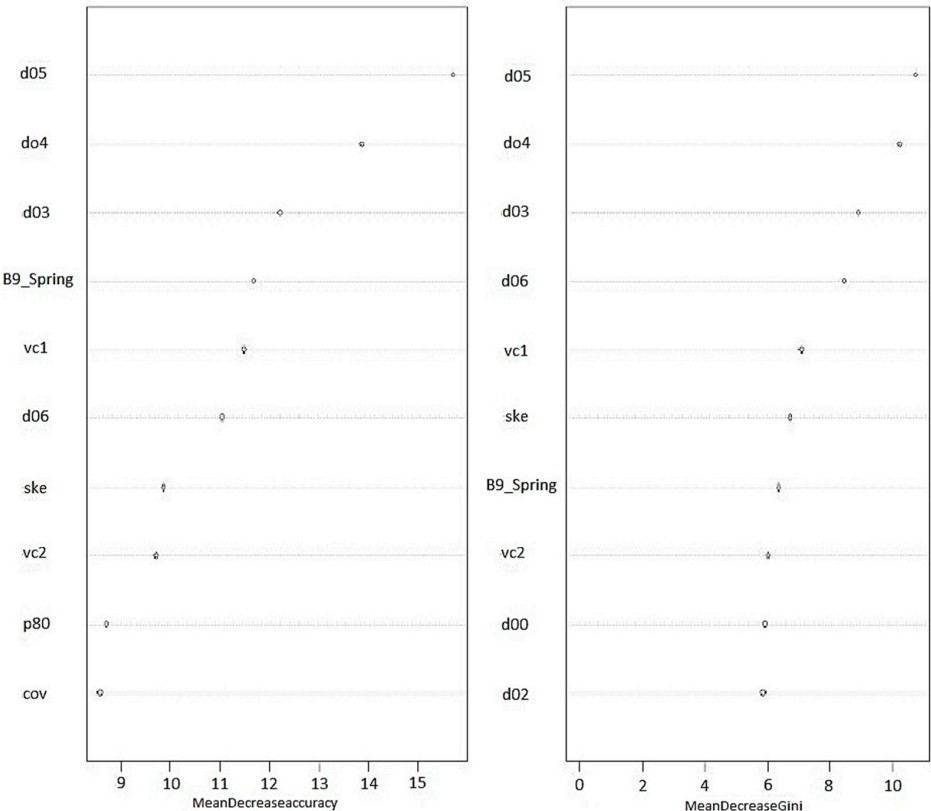

**Figure 7.** The ten most important variables from RF model in terms of Mean Decrease Accuracy (**left**) and Mean Decrease in Gini (**right**).

Finally, maps of 13 fuel sub-fuel models for the Serra da Lousã and Mafra study areas at 25 m of resolution were created using routines with the *terra* package in R software for the study areas (Figure 8).

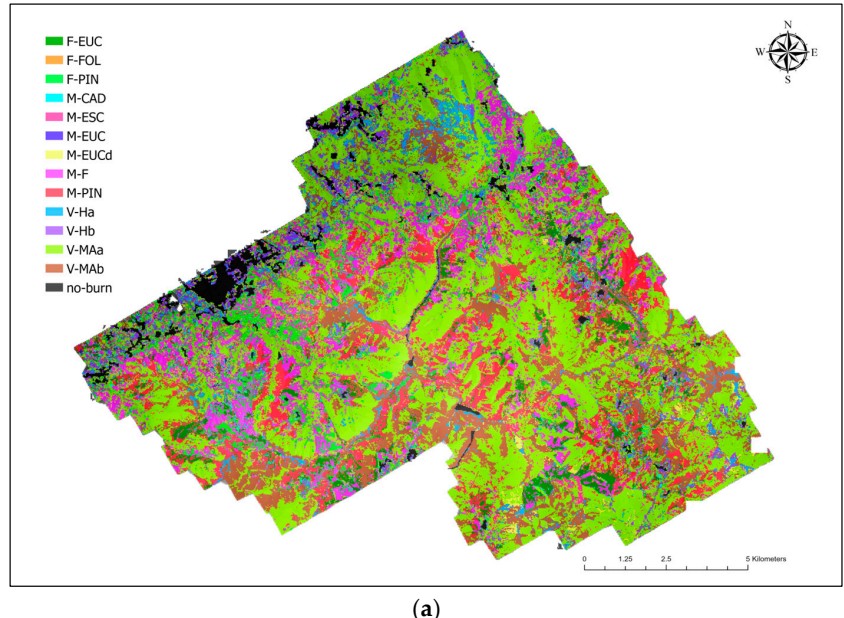

(**a**)

**Figure 8.** *Cont.*

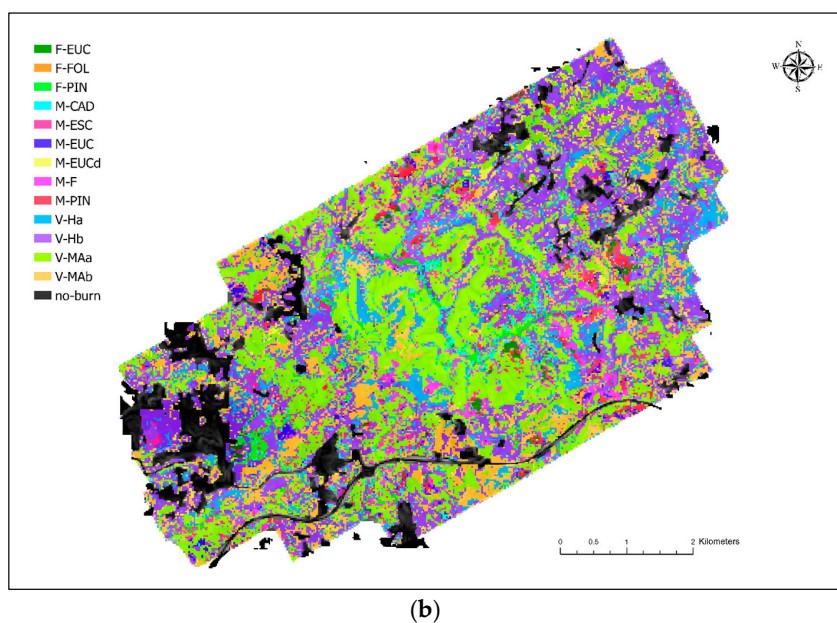

(**b**)

**Figure 8.** Maps of fuel models in Portugal for the Serra da Lousã (**a**) and Mafra (**b**) study areas.

## 4. Discussion

This study investigated the usefulness of medium point density ALS data, S1, S2 and PALSAR data to classify and map fuel types in six topographically varied areas, with complex land cover, in Portugal. The models used quantified the relationship between ALS and field-based measurements, assessing the effects of canopy cover and of pulse density. We also assessed the performance of multi-source sensors for classifying a large number of fuel types, combining ALS, S1, S2, and PALSAR data.

Regarding the first objective of the study, fuel model classification using CART with ALS variables only, the results indicate that understory vegetation in Portugal can be classified with moderate accuracy (Table 7) into the D, F, M and V fuel groups, with UA values of 0.50, 0.81, 0.63 and 0.70, respectively. The most problematic fuel model group was Discontinuous fuel (D), with an estimated UA of 0.50 and a PA of 0.11. A possible reason for this is classification of the fuel based on the system used in Portugal leads to some confusion between similar fuel models, resulting in the inaccurate classification of class D. To improve accuracy, fuel model D would have to be eliminated and its fuel models distributed by the other three fuel model groups. ALS variables, particularly relative density and vertical complexity index, were found to be the most important for classifying understory fuel models. The use of CART, simple CART and RF methods in this analysis has demonstrated the importance of each method for specific applications. The RF method yielded the poorest results, with an OA of 0.60 and a kappa value of 0.34, whereas CART produced the best results in this specific application. Among the three classification approaches examined, the RF model had poor accuracy, which is consistent with the findings of previous studies with similar objectives [62,63]. The results confirmed the poor performance of RF, probably due to the small sample sizes for some fuel model groups. The study findings also indicated that the moderate results may be due to the limitation of the discrete ALS sensor used, which may provide more biased and less consistent measurements of forest understory structure than the full waveform ALS [64–66]. It is possible that our results may be improved at finer scales by using drone-LiDAR or terrestrial LiDAR scanning (TLS) [67,68], although each method has its own limitations and cover smaller areas than ALS.

Our study used the three main canopy cover (CC) groups considered in the Portuguese NFI to analyze the effect of the number of plots per classification group of the model. A study in semi-arid conifer-dominated forests in the southwestern USA [69] concluded that canopy cover can be used as a proxy for stand density when developing a combined individual tree distribution with area-based approaches for estimating understory. In our

study, better results were obtained for Open Forest than for Dense Forest CC, with an OA of 0.75 and a kappa value of 0.57 (Table 10). The correlation between field measurements and the ALS-derived structural characteristics of ground and understory vegetation depends on the forest type and the ALS data configuration. Such values may be different in forests with more closed canopies or sparser ALS point density [70]. These findings highlight the importance of tree canopy cover for fuel model classification accuracy.

Even though ALS pulse density is considered an important factor in relation to understory estimation and fuel type classification, the results reveal that the models yielded better results with higher point density (10 points $m^{-2}$) than with lower point density (5 points $m^{-2}$). The best results were obtained using CART without cross validation, with 0.71 OA for 5 points $m^{-2}$ data and 0.78 for 10 points $m^{-2}$ data. Better results were obtained with a higher point density given that those values are based on almost twice the number of plots (316 plots) than with the lower point density (183 plots). We suspect that an imbalanced number of plots within the point density groups may also have hurt classification results. Previous studies produced good estimates using an even lower point density. For instance, a study that compared two sets of data with densities of 0.5 and 2 points $m^{-2}$ [46] produced non-significant improvements with an increasing pulse density for vegetation structure estimates based on the Prometheus classification scheme. In a study using six different datasets with point densities ranging from 0.5 to 10 points $m^{-2}$ [71], the authors found that accurate estimates of vertical canopy structure can be obtained even with a pulse density of 0.5 points $m^{-2}$ across all forest types. However, the results are not directly comparable since we are classifying understory fuel model with variables related with vertical canopy structure.

Regarding the multisensor approach, the results obtained with the RF model yielded a moderate OA of 0.44 and a kappa coefficient of 0.31 (Table 14). The results of the present study cannot be compared directly with those of other fuel mapping studies as different methodologies were used, as was a different classification scheme specifically adapted for fuel types in Portugal. In a study combining ALS data with Landsat 9 imagery, researchers obtained an OA of 0.82 and a kappa value of 0.77 by using two different fuel classification schemes—Northern Forest Fire Laboratory (NFFL) and the specific Canary Island fuel model (CIFM) [48]. Regarding the derived variables, our findings can be compared with those of most other studies using ALS data for fuel mapping. However, we used spectral index variables from different seasons (spring, summer, and autumn). While the four main fuel groups in Portugal were classified with reasonable accuracy, RF yielded poor results for 13 specific fuel models. The RF models performed similarly to other studies [17], which obtained slightly better results, with an OA of 0.59 for seven fuel models based on the Prometheus classification. Therefore, better performances are expected for fewer fuel models [17,40], increased point density [36], and higher-resolution Multispectral or even Hyperspectral and UAV (Unmanned Aerial Vehicle)-LiDAR imagery [72]. Even considering the good accuracy observed in other studies using C-SAR bands [33,39,73] to classify land-cover vegetation fuel models, the present findings suggest that the use of SAR variables did not improve the accuracy of the classification of fuel model types, with an OA of 0.30 and a kappa of 0.17. One of the possible reasons is the low capacity of the signal to penetrate through vegetation of L-band SAR and C-band SAR from S1 and PALSAR2. Topography and variables derived from S2 data were found to be the most important for classification, rather than L and C SAR bands. Future improvements may be obtained in order to classify fuel model types with the upcoming NASA-ISRO Synthetic Aperture RaDaR (NISAR) satellite mission in 2024 that will deliver denser L-band time series data at a higher spatial resolution of 12 m [74].

The results of the present study could be also explained by the imbalanced fuel model observations in our study; in such cases, the model may struggle to learn patterns from the minority fuel model due to limited data, and it might end up biased towards the majority fuel models. Several authors have faced similar problems with different numbers of observations [75,76]. However, it is worth mentioning that most studies in the literature

used fewer fuel models and plots [17,46,77]. For all of the metrics together (ALS + S1 + S2 + PALSAR + DEM), the results showed that the first 20 most important variables include non-texture variables from PALSAR and S1, and PALSAR variables were more important than the variables from S1 (Figure A4, Appendix B). Findings of similar studies suggest that the S1 satellite RaDaR images have a shorter wavelength than the ALOS/PALSAR and consequently low canopy penetration [78]. One interesting finding is the great improvement in including ALS data with the RaDaR variables, and that this increases the OA to 0.41 and the kappa value to 0.32. In contrast to ALS variables, optical metrics and spectral indices derived from S2 do not provide information on vegetation structure, but rather, they describe the photosynthetic activity of plants [79]. One of the possible reasons for the results obtained is that fuel models may involve different species in the same group (for example, M-CAD and M-PIN) with different spectral signals. This may be attributed to the high number of mixed forest training plots with high species heterogeneity. Regardless of these limitations, data derived from the S2 optical sensor proved to be more useful than those from ALOS/PALSAR and S1.

An important issue in terms of analysis is also the different number of variables from different seasons obtained with S2. As the results showed, summer and spring variables include spectral indices that were ranked highly in terms of importance (Figure A4, Appendix B). In addition to the seasonality, the confusion matrix indicated difficulties in distinguishing between litter with fern understory (M-F) and tall shrubs (V-MAa). As the M-F fuel model has a very similar plant structure to the shrubs (V-MAa), the RF model had some difficulty in identifying and distinguishing these two fuel models. Litter of intermediate to long needle pines (F-PIN) and Litter of intermediate to long needle pines and shrub under-story (M-PIN) were also similar. The model classified nine correct observations of F-PIN and predicted seven observations as M-PIN (Table 15), although these two fuel models are similar in terms of structure in the overstory but with different understory vegetation. The most promising results were obtained to classify tall- and low-shrub fuel models (V-Ma and V-Mb). In terms of producer accuracy, V-Ma and V-Mb yielded PAs of 0.69 and 0.52, respectively. Fuel models M-F and V-Ha also yielded good PAs of 0.55 and 0.60, respectively.

Improvements in future fuel mapping studies could be made by focusing on some key points. One way would be to develop fuel models less variable in structure and composition. Improvement could also be achieved by increasing the number of plots sampled and their representativeness for each fuel model. Finally, focusing on creating a more balanced sampling design would prevent large variations in the number sample plots by fuel model under analysis.

## 5. Conclusions

This study focused on using RS data to classify fuel model groups and map fuel models in six environmentally diverse areas in Portugal. The aim was to provide a representative and accurate classification of fuel model groups at the regional scale to help in forest management and fire behavior simulation. This study used medium point density ALS data and S1, S2, and PALSAR2 data to classify and map fuel models. The results of the study showed that data from ALS sensors can yield a moderately accurate classification of fuel model groups at a regional scale within the study areas. The CART method produced very good results in Sparse forests and correctly classified the four main fuel groups in Portugal. The study also showed that the LiDAR point density effect on the final classification of the fuel model using the CART method with 10 points m$^{-2}$ data produced better results than with the 5 points m$^{-2}$ density data. The multi-sensor approach showed that the use of SAR variables derived from S1 and PALSAR satellites did not improve the model, and ALS alone associated with DEM variables achieved the same performance.

Further research is warranted to improve the understanding of understory forest structure, focusing on ALS sensors and their spatial and temporal resolution. Future research should ideally include a ALS coverage for the entire country of Portugal, increase

the number of observations per fuel model, and rethink fuel model development in terms of structure and composition. The effects of using High-Resolution Multispectral Imagery (HRMSI) and Hyperspectral Imagery (HSI) associated with spectral indices should be compared with the aim of creating a more balanced sample design to avoid the high variations in the number of sample plots under analysis. The study findings may be important for fuel treatment planning and for fire behavior simulation.

**Author Contributions:** Conceptualization, B.M. and J.G.-H.; methodology, B.M., P.M.F., J.M.C.P. and J.G.-H.; software, B.M. and J.G.-H.; validation, B.M. and J.G.-H.; formal analysis, B.M., P.M.F., J.M.C.P. and J.G.-H.; investigation, B.M. and J.G.-H.; resources, B.M., P.M.F. and J.G.-H.; data curation, P.M.F. and J.M.C.P.; writing—original draft preparation, B.M. and J.G.-H.; writing—review and editing, P.M.F. and J.M.C.P.; visualization, B.M. and J.G.-H.; supervision, P.M.F., J.M.C.P. and J.G.-H.; project administration, J.G.-H. and P.M.F.; funding acquisition, J.G.-H. and P.M.F. All authors have read and agreed to the published version of the manuscript.

**Funding:** This research was supported by the grant funded by the Foundation for Science and Technology (FCT), Portugal to Guerra-Hernández(#CEECIND/02576/2022). This work was also supported by the European Horizon 2020 research and Innovation Programme under grant agreement nº 101037419, through a project called FIRE-RES–Innovative technologies and socio-economic solutions for fire-resistant territories in Europe. JMC Pereira and P Fernandes participations were supported by Fundação para a Ciência e a Tecnologia I.P. (FCT), respectively through projects UIDB/00239/2020 and UIDB/04033/2020.

**Institutional Review Board Statement:** Not applicable.

**Informed Consent Statement:** Not applicable.

**Data Availability Statement:** The datasets analyzed for this study are available from the corresponding author upon reasonable request.

**Acknowledgments:** The ForestWISE–Collaborative Laboratory for Integrated Forest and Fire Management is acknowledged for making available the ground data and the enhanced plot positioning information of the project áGiL.

**Conflicts of Interest:** The authors declare no conflict of interest.

## Appendix A

**Table A1.** Area calculated by Google Earth Engine (GEE) in ha associated with elevation range from Digital Elevation Model (DEM) for each study area.

| Study Area | Area Calculated (ha) by GEE | Elevation Derived from DEM (m.a.s.l) |
|---|---|---|
| Mafra | 2278 | 0–357 |
| Pombal | 5103 | 0–347 |
| Monsanto | 1210 | −10.98–208 |
| Sintra Cascais | 8156 | −20.67–520 |
| Serras da Lousã | 11,784 | 0–1.203 |
| Vila Pouca de Aguiar | 5578 | 207.51–1196 |
| Total: | 34,109 | |

**Table A2.** Portuguese fuel models and description.

| Fuel Group | Fuel Model | Description |
|---|---|---|
| LITTER | | |
| F | F–RAC | Compact conifer litter of short-needled pines |
| F | F–EUC | Eucalypt litter |
| F | F–FOL | Compact litter of deciduous or evergreen hardwood |
| F | F–PIN | Litter of medium to long needle pines |

**Table A2.** *Cont.*

| Fuel Group | Fuel Model | Description |
|---|---|---|
| LITTER AND VEGETATION | | |
| M | M–CAD | Deciduous hardwood litter and shrub understory |
| M | M–ESC | Evergreen hardwood litter and shrub understory |
| M | M–EUC | Eucalypt litter and shrub understory |
| M | M–EUCd | Discontinuous surface fuels in eucalyptus plantations |
| M | M–PIN | Litter of medium to long-needled pines and shrub understory |
| M | M–H | Litter with herbaceous understory |
| M | M–F | Litter with fern understory |
| VEGETATION | | |
| V | V–Ha | Tall grass (>0.5 m) |
| V | V–Hb | Low grass (<0.5 m) |
| V | V–MAa | Tall Shrubs (>1 m) with substantial fine and or dead fuel |
| V | V–MAb | Low Shrubs (<1 m) with substantial fine and or dead fuel |
| V | V–MMb | Low Shrubs (<1 m) poor in fine dead fuel |
| V | V–MMa | Tall Shrubs (>1 m) poor in fine dead fuel |
| V | V–MH | Low Shrubs (<1 m), poor in dead fuel and discontinuous, often mixed with grass |

**Table A3.** Sentinel 2 predictor variables associated with vegetation indices computed for summer, winter, and spring seasons. Sentinel 1 Backscatter H and V polarization variables associated with GLCM metrics computed for the period 1 June 2020 to 30 July 2020. ALOS2/PALSAR2 Backscatter H and V polarization variables with GLCM metrics predicted variables extracted for yearly mosaic.

| Sensor | Predictor Variables | Description |
|---|---|---|
| SENTINEL-2 AND VEGETATION INDICES | B2 | Blue |
| | B3 | Green |
| | B4 | Red |
| | B5 | Vegetation Red Edge |
| | B6 | Vegetation Red Edge |
| | B7 | Vegetation Red Edge |
| | B8 | Near Infrared (NIR) |
| | B8a (B9) | Vegetation Red Edge |
| | B11 | Short Wave Infrared (SWIR) |
| | B12 | Short Wave Infrared (SWIR) |
| | NDVI | Normalized Difference Vegetation Index |
| | OSAVI | Optimized Soil Adjustment Vegetation index |
| | GNDVI | Green Normalized Difference Vegetation Index |
| | NDWI | Normalized Difference Water Index |
| SENTINEL-1 | Backscatter GLCM metrics | VV and VH |
| | | VV_Mean |
| | | VV_Variance |
| | | VV_Homogeneity |
| | | VH_Mean |
| | | VH_Variance |
| | | VH_Homogeneity |
| ALOS2/PALSAR | Backscatter GLCM metrics | HV and HH |
| | | HV_Mean |
| | | HV_Variance |
| | | HV_Homogeneity |
| | | HH_Mean |
| | | HH_Variance |
| | | HH_Homogeneity |

## Appendix B

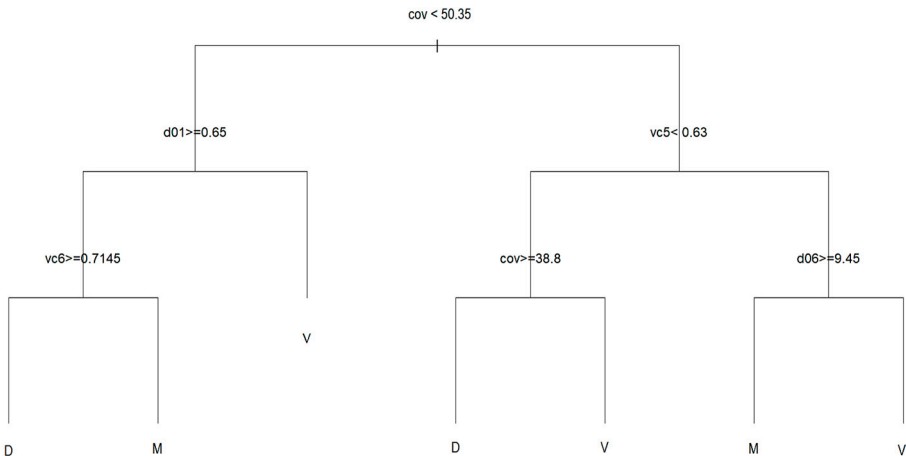

**Figure A1.** Results of CART model of the classification Open Forest.

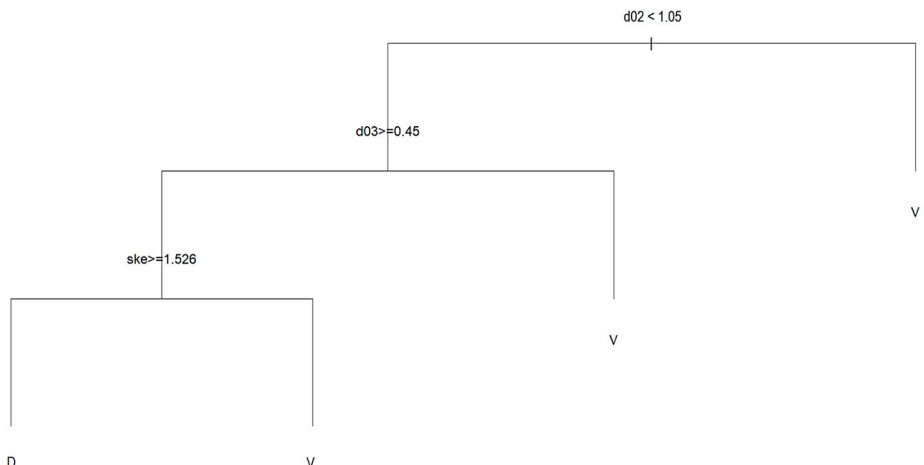

**Figure A2.** Results of CART model of the classification Sparse Forest.

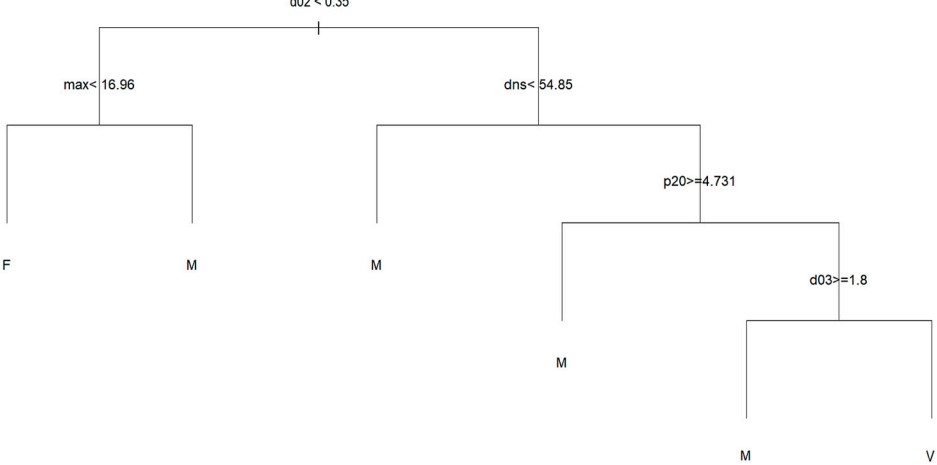

**Figure A3.** Results of CART model of the classification Dense Forest.

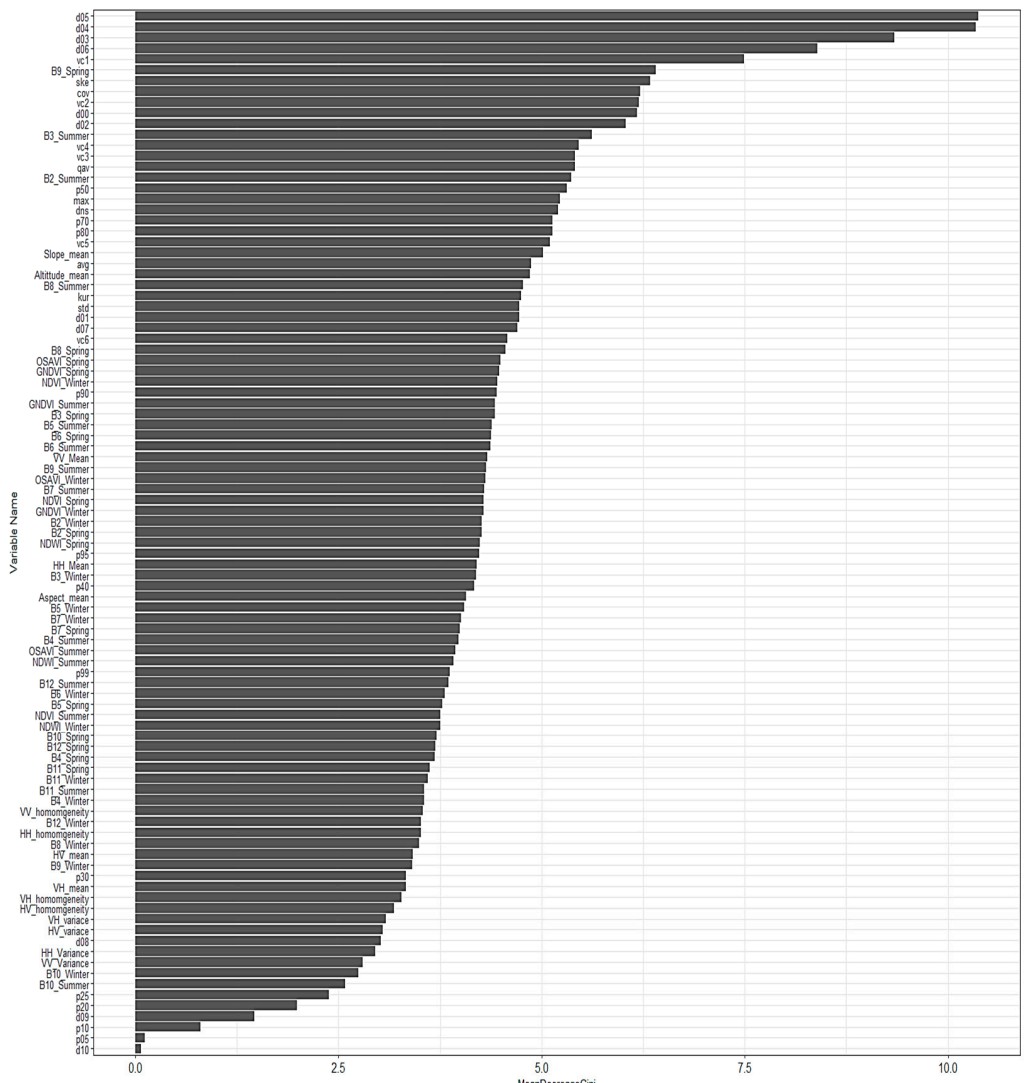

**Figure A4.** Results of the most important variables of RF using multisensory approach.

**Table A4.** CART without cross validation for Sparse Forest Classification accuracy for different fuel groups, where D = discontinuous fuel, F = Litter, M = Mixed, V = Vegetation, PA = Producer's accuracy, UA = user's accuracy, and value shown in bold OA = overall accuracy and kappa value. Classification and reference (field check) frequencies are arranged in columns and rows, respectively.

| | | Observed | | | | | |
|---|---|---|---|---|---|---|---|
| | Fuel group | D | F | M | V | Σ | PA |
| Predicted | D | 14 | 2 | 0 | 3 | 19 | 0.74 |
| | F | 2 | 0 | 0 | 4 | 6 | 0.00 |
| | M | 2 | 0 | 0 | 5 | 7 | 0.00 |
| | V | 6 | 0 | 0 | 109 | 115 | 0.95 |
| | Σ | 24 | 2 | 0 | 121 | 147 | |
| | UA | 0.58 | 0.00 | 0.00 | 0.90 | OA | 0.84 |
| Kappa = 0.51 | | | | | | | |

**Table A5.** CART without cross validation for Open Forest Classification accuracy for different fuel groups, where D = discontinuous fuel, M = Mixed, V = Vegetation, PA = Producer's accuracy, UA = user's accuracy, and value shown in bold OA = overall accuracy and kappa value. Classification and reference (field check) frequencies are arranged in columns and rows, respectively.

|  |  | Observed | | | | |
|---|---|---|---|---|---|---|
|  | Fuel group | D | M | V | Σ | PA |
| Predicted | D | 13 | 2 | 7 | 22 | 0.60 |
|  | M | 2 | 27 | 11 | 40 | 0.68 |
|  | V | 5 | 6 | 60 | 71 | 0.85 |
|  | Σ | 20 | 35 | 78 | 133 |  |
|  | UA | 0.65 | 0.77 | 0.77 | OA | 0.75 |
| Kappa = 0.57 | | | | | | |

**Table A6.** CART with cross validation for Dense Forest Classification accuracy for different fuel groups, where D = discontinuous fuel, F = Litter, M = Mixed, V = Vegetation, PA = Producer's accuracy, UA = user's accuracy, and value shown in bold OA = overall accuracy and kappa value. Classification and reference (field check) frequencies are arranged in columns and rows, respectively.

|  |  | Observed | | | | | |
|---|---|---|---|---|---|---|---|
|  | Fuel group | D | F | M | V | Σ | PA |
| Predicted | D | 4 | 3 | 17 | 5 | 29 | 0.18 |
|  | F | 0 | 22 | 14 | 4 | 40 | 0.55 |
|  | M | 2 | 2 | 73 | 10 | 87 | 0.84 |
|  | V | 2 | 3 | 13 | 45 | 63 | 0.71 |
|  | Σ | 8 | 30 | 117 | 64 | 219 |  |
|  | UA | 0.50 | 0.73 | 0.62 | 0.70 | OA | 0.66 |
| Kappa = 0.50 | | | | | | | |

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
