# Peer review of "Comparing Forest Understory Fuel Classification in Portugal Using Discrete Airborne Laser Scanning Data and Satellite Multi-Source Remote Sensing Data"

_fire, doi:10.3390/fire6090327_

Round 1

Reviewer 1 Report

The writing is strong and the subject matter is fascinating. A scientific analysis of the research problem is conducted. The current study focused on classifying and mapping fuel types using remote sensing (RS) data in six topographically diverse areas of Portugal. This will aid in forest management and fire detection throughout the nation.

To categorize and map different fuel types, the author used medium point density ALS, S1, S2, and PAL-SAR2 data. Although not more recent, this methodology was effective for this study. For the study, they employed the CART methodology. To make the results more comparable and practical, the author can provide information about the understory of the forest structure of some other significant fuel tree species.

The analysis of temporal data can also be added to the study to provide estimates for future predictions. To manage fire safety, it will be useful to evaluate the forest areas in Portugal.

To express the validity of the result, the author needs to include some supporting references in the conclusion section. It is advised that the author include some references from more recent years.

 Minor editing of English language required

Reviewer 2 Report

General Comments

The manuscript entitled “Comparing Fuel Classification in Portugal by Using the National Classification System, Discrete ALS Data and Multi-source Remote Sensing Data” aims to classify the main fuel types in Portugal, focusing on the horizontal and vertical structure, based on field data obtained by 499 plots distributed in 6 different areas of Portugal. To do this, several CART and Random forest models were developed. For the classification, the canopy cover effect and the use of different ALS pulse densities were investigated. Another aim of the work was the mapping of a more complex fuel type classification by the use of a multi-sensor approach and the RF method.

Overall, the manuscript is interesting and ambitious and aims to represent a further step in the knowledge of fuel conditions and fuel type mapping. In this sense, the value of the paper is high. However, there are some parts that should be improved in order to increase the quality and clarity of the manuscript. The English form needs some improvements in some parts of the manuscript. Several punctuation errors are present, and decimals and thousands are often misspelled. Many citations are not numbered. Moreover, since I do not know if it is an error of the authors, I point out that after Table 14, the number of lines starts again, while in L368 there is a second Table 8. In addition, the Results section is not mentioned in the text; I suppose it starts from L318.

The Introduction section provides an exhaustive overview of the state of the art and is in my opinion fine, except for a few parts that are provided in the section “specific comments”. The Material and Method section is too long and should be better organized in order to focus on the most important steps of the work. Despite six study areas being presented, only four of them are shown in the maps of the paper.

Also, in the Result section, Tables are too numerous: if possible, I would suggest the authors merge some of them (e.g. tables 6, 7, 8 (line 300)) to make their reading and the comparison of results easier. Instead, I find that the number of maps presented in this work is small, whereas, in a work like this, they could be of great importance, especially for the presentation of the results.

Overall, a minor revision of the work before publication in the Fire journal is needed.

Specific comments

L46-57: Here, the description of the most affected fuel types in the Mediterranean basin is poor. I suggest enriching the bibliography and adding some other references

L51: I think that here, the description of the Portuguese vegetation is not necessary. Instead, a better description of the study areas would be appreciated

L56-70: Perhaps, this paragraph requires some more accurate description of the main methods used for fuel type classification, together with the works that applied that approaches (including some of them based on field surveys)

L98 Domingo et al needs the citation number

L132 usually, a comma must be used to separate miles, while a dot for decimal units

L136 I suggest including a general description of the study areas, and explaining the choice for the selected areas

L138 in Figure 1 only the land cover maps for 4 of the 6 study areas are shown. On the upper side of the map, please specify to which study area the row Total is referred to

L150 please add the references for the fuel classification system used in Portugal

L157 Table1 please, in the caption better explain the table, together with the meaning of the colors used

L191 Table2 I suggest reorganizing the table because I've found it difficult to understand

L213 if the column temporal aggregation is the same for all the rows, I suggest deleting it from the table and providing this information in the text

L318 since it is not specified, I suppose the Results section starts here

Fig.8 and 9, why only 2 of 6 study areas are shown in the maps?

Reviewer 3 Report

Overall, the study is very interesting and can have a lot of applicability to modeling fuel bed types/fuels across the world and not just in Portugal. The comparisons of different models (CART with and without cross-validation) with Random Forest are also very interesting, along with the comparison of ALS using different pulse rates. In general, I think the premise of this study would be very interesting to the field as a whole and could be of interest to publish.

However, there are some significant errors in the Results section. Frequently, the values cited in the text did not match the data in the tables. It unclear if the data in the tables is incorrect, the data in the text is incorrect, or both are incorrect. The merits of this manuscript and its conclusions cannot be recommended for publication until the Results section is thoroughly and carefully reviewed by the authors and corrected, and the any conclusions are confirmed to be based off the true results.

In addition, there are some areas where further discussion is needed. In the Discussion section, some variables are discussed in terms of their importance to the model (i.e. NDVI, NDWI, non-texture variables from PALSAR and S1), but these rankings of the model variables do not appear in any table in the main text or supplement. For example, the Discussion talks about the first 20 most important variables for a model, yet these 20 variables are not shown. This information should be included if it is discussed in the paper.

Overall:

·         For numbers, numbering conventions are used interchangeably and sometimes mixed together. For example, line 132 uses 34.109 ha, which I assume the equivalent would be 34,109 ha in English mathematics/literature, but line 134 uses 1.203.48 m.a.s.l. I assume this is either 1,204.48 m.a.s.l. or 1.204,48 m.a.s.l., but it is unclear. This happens a few other times throughout the manuscript. I am not sure what the journal convention is, but the convention should be uniform throughout the entire paper.

·         Can the best/final model diagrams be shown for all models like in figures 4 and 5? These can be included in the supplement. In particular, for sparse, open, and dense forests?

Particular:

·         Line 59: add in a closed parenthesis to match open parenthesis on line 58. On line 59, it should read “…amongst others).

·         Line 63: remove the word “relative” from “…is used [15], relative compared to…”

·         Line 65: BEHAVE had not been defined previously. Define the acronym and include citation.

·         Line 111: remove “a” from “On the other hand, a very few studies….”

·         Line 148: remove parenthesis before “45 degrees” and replace with the word “at” to read “…were defined at 45 degrees…”

·         Line 154: Are these numbers supposed to match Table 2 (the first Table 2 on line 168)? The values in the text and table do not match (F is 46 plots in text vs. 60 in table, M is 134 vs 173, and V is 249 vs 242)

·         Line 191: This table is also listed as Table 2, but there is already a Table 2 on Line 168. This table is also difficult to read. Perhaps removing the underline from 1.1, 1.2, 1.3, etc would help

·         Line 192: This is a fantastic figure, but it is never cited or discussed in the text. I believe some discussion of this figure takes place before line 191, but cite the figure in the text and include a more robust discussion of what is being shown in the figure.

·         Equation 1: the font makes the l in ln look like a capital i

·         Line 202: a comma is used where a period is used in the other numbers

·         Line 209: remove “before being processed”

·         Line 212: This is referencing Table 3 in the text (although the Table numbers are off)

·         Line 213: This is Table 3 in the text, but is referred to as Table 4 on line 212

o   Also, should the “Sensor” heading be over “S2”?

·         Line 229: Explain to the reader how using the backscatter values help estimate understory vegetation. What is this step? What is exactly in the matrices that are derived?

·         Lines 232-234: All of these variables should be listed somewhere as some are discussed later in the text. Include a supplement table that lists each variable, what sensor it is from, and how it is calculated for all variables not already described in the main text such as in Table 3 (line 213).

·         Line 245: remove “to growth”

·         Line 248: add a comma after “Producer (PA)”  

·         Line 259: There is a missing section heading for Section 3. I think it should be, “Section 3. Results”

·         Line 269: add that this is the CART method with cross-validation as there are two different CART methods being used.

·         Line 276/Figure 4.

o   While it is stated that this model can classify for D, D does not appear in the model diagram so how can the model classify anything as D? Is this the correct model diagram?

o   Although they may be defined elsewhere, define the abbreviations used (i.e. cov, ske, d02, etc.) or refer the reader to the table where they are defined.

o   In the figure caption, include what model this diagram is for as there are many models and diagrams in this paper. For example, “CART fuel classification model using ALS data”

·         Line 278/Table 6

o   As there are several similar tables in this manuscript, put in table caption that this is for “CART fuel classification model using ALS data”

·         Line 288: text and table mismatch between class M (text says 63, Table 7 lists 84) and class V (text says 226, Table 7 says 200).

·         Line 294/Table 7

o   Either the table is incorrect, or the text above on Line 288 is incorrect

·         Line 297: Table 8 lists PA value of 0.06, text says PA value of 0.17. Is this supposed to be UA value as that is 0.17, or is it supposed to be the PA value of 0.06? Or is the table incorrect?

·         Line 308/Figure 6:

o   The “max” value also seems significant, why not discuss that value?

o   Why not discuss the results from “MeanDecreaseGini”? If it is not discussed, it should not be included in the main text, or it can be listed as a supplement for reference.

·         Line 324: I believe these results come from Table 9. Cite that in the text

·         Line 331: remove semi-colon

·         Line 333: Cite that the results come from Table 9

·         Lines 340-344: several kappa and OA values do not match between what is stated in the text, main text Table 9, and supplement Table B3

·         Line 344: cite values came from Table 9

·         Line 351/Table 10: round values to 2 decimal places as this is what is shown in the text

·         Line 356: Text says Table 16, but it should be Table 11

·         Line 371: Table gives PA accuracy of 71%, but table 12 shows 88%

·         Line 372: PA should be 84%, not 0.84%

·         Line 377/Table 12: There is a mismatch between PA values cited in line 372 and in Table 12.

·         Line 381: Include a supplement table that lists all variables used

·         Lines 391-418

o   There were many difference between what was stated in the text and given in the tables so it is unclear which is correct. Here are the differences

§  Line 398: highest PA in Table 14 is F-EUC at 0.42

§  Line 400: M-F produced the highest PA followed by M-ESC unless these are based off of UA values? It is unclear which metric is being used

§  Line 407: remove the “m” after 0.37

§  Lines 408-413: Is this reversed? The table shows the model predicted M-F when the observed landcover was V-MAa 13 times. The table shows the model predicted V-MAa when it was actually observed to be M-F 12 times. Is the text or table incorrect?

§  Lines 413-414: The sentence starting with “Interestingly, the results estimated …” is incomplete and unclear.

§  Line 418: V-Ha and V-Hb values in the text and table do not match

·         Line 419/Figure 7: Make font bigger. It is difficult to read.

·         Table 14: List that this is for the RF model

Line renumbering occurred after Table 14 – not sure why.

·         Line 2 after Table 14:

o   Should be figure 7, not figure 8

·         Line 6 after Table 14: I do not see slope and altitude on Figure 7. Where is this data from?

·         Line 7 after Table 14: Discuss the sentence that starts here in more detail. What variables belong to each of these sensors? You can even color code the variables in Figure 7 to match the sensor. For example, ALS-derived variables are in blue, S2 are in red, etc.

·         Include landcover figures for the other areas in the supplement.

·         Line 24 after Table 14: There is no Table 5. Is this supposed to be table 6?

·         Line 25 after Table 14: The UA values in the text do not match the table

·         Lines 44-45 after Table 14: The plot number values do not match Table 8 (147, 135, and 217, respectively)

·         Line 47 after Table 14: note that this is the CART method without cross-validation as with and without cross-validation were performed.

·         Line 60 after Table 14: Tables 11 and 12 list different values than those given in text (246 and 130, respectively)

·         Line 96 after Table 14: note that the OA and kappa values are from Table 13

·         Line 100 after Table 14: Show a table that ranks the 20 most important variables from PALSAR and S1 as these are discussed in the paper, but they are not shown anywhere.

·         Line 114 after Table 14: There is discussion that NDVI and NDWI were ranked high in variable importance, but the variable rankings are not shown. Show these in a supplement.

·         Line 117 after Table 14: List the different variables from the different seasons obtained with S2

·         Line 124 after Table 14: Table 14 says there were 9 correct observations, but text says 8

·         Appendix Table B3: Kappa value is not what is listed in the text

There are some differences in how numbers are shown (i.e. 4.200 ha vs. 4,200 ha) and sometimes the conventions are mixed. 

The English language usage is clear with no issues.

Reviewer 4 Report

The manuscript is interesting and fits the scope of the journal. I have some suggestions for the authors to improve the paper.

(i) The acronym should be explained the first time they appear. Then the authors could simply use the acronym.

(ii) The references in the text should be reported by means of square brackets, as required by the style of the journal.

(iii) The numbering of the tables should be revised since there are some problems with them.

(iv) To properly understand Eq.(2) – Eq.(5), it is necessary explaining their parameters.

(v) I believe that there are some problems with the numbering of the sections: I do not find the section 3. Moreover, the main current problem with the manuscript is the integration between methods and results in an only section: the readers should distinguish between the methodology and the results obtained thanks to it. The authors should therefore separate the proposed method from the main outcomes of their approach.

Round 2

Reviewer 3 Report

I appreciate the authors taking time to read my comments and incorporate them into their article. The discussion is more robust and provides a lot of good information for the community. I believe that this research will be of interest to anyone wanting to use ALS and drones in the future to map fuels and fuel structure. 

I have just a few comments on things to correct, but they are very minor.

Pg. 8, line 206: change from 0,5 m (vc2) to 0.5 m (vc2)

Pg. 15, line 358: The tables for the confusion matrixes are B1, B2, and B3, not B2, B3, and B4

Thank you for conducting this research and publishing it.

Author Response

Respected, please see the attachment. 

Best regards,

Bojan Mihajlovski 
